# M Segment-Based Minigenome System of Severe Fever with Thrombocytopenia Syndrome Virus as a Tool for Antiviral Drug Screening

**DOI:** 10.3390/v13061061

**Published:** 2021-06-03

**Authors:** Hiroshi Yamada, Satoshi Taniguchi, Masayuki Shimojima, Long Tan, Miyuki Kimura, Yoshitomo Morinaga, Takasuke Fukuhara, Yoshiharu Matsuura, Takashi Komeno, Yousuke Furuta, Masayuki Saijo, Hideki Tani

**Affiliations:** 1Department of Microbiology, Faculty of Medicine, Academic Assembly, University of Toyama, Toyama 930-0194, Japan; hiyamada@med.u-toyama.ac.jp (H.Y.); tanl_0706@163.com (L.T.); mkimura@med.u-toyama.ac.jp (M.K.); morinaga@med.u-toyama.ac.jp (Y.M.); 2Department of Virology I, National Institute of Infectious Diseases, Tokyo 162-8640, Japan; rei-tani@nih.go.jp (S.T.); shimoji-@nih.go.jp (M.S.); msaijo@nih.go.jp (M.S.); 3Department of Molecular Virology, Research Institute for Microbial Diseases, Osaka University, Osaka 565-0871, Japan; fukut@pop.med.hokudai.ac.jp (T.F.); matsuura@biken.osaka-u.ac.jp (Y.M.); 4Department of Microbiology and Immunology, Graduate School of Medicine, Hokkaido University, Hokkaido 060-8638, Japan; 5Center for Infectious Diseases Education and Research (CiDER), Research Institute for Microbial Diseases, Osaka University, Osaka 565-0871, Japan; 6FUJIFILM Toyama Chemical Co., Ltd., Toyama 930-8508, Japan; takashi.komeno@fujifilm.com (T.K.); yousuke.furuta@fujifilm.com (Y.F.); 7Department of Virology, Toyama Institute of Health, Toyama 939-0363, Japan

**Keywords:** SFTSV, minigenome assay, antivirals, antiviral screening, favipiravir, ribavirin

## Abstract

Severe fever with thrombocytopenia syndrome virus (SFTSV) is an emerging tick-borne bunyavirus that causes severe disease in humans with case fatality rates of approximately 30%. There are few treatment options for SFTSV infection. SFTSV RNA synthesis is conducted using a virus-encoded complex with RNA-dependent RNA polymerase activity that is required for viral propagation. This complex and its activities are, therefore, potential antiviral targets. A library of small molecule compounds was processed using a high-throughput screening (HTS) based on an SFTSV minigenome assay (MGA) in a 96-well microplate format to identify potential lead inhibitors of SFTSV RNA synthesis. The assay confirmed inhibitory activities of previously reported SFTSV inhibitors, favipiravir and ribavirin. A small-scale screening using MGA identified four candidate inhibitors that inhibited SFTSV minigenome activity by more than 80% while exhibiting less than 20% cell cytotoxicity with selectivity index (SI) values of more than 100. These included mycophenolate mofetil, methotrexate, clofarabine, and bleomycin. Overall, these data demonstrate that the SFTSV MGA is useful for anti-SFTSV drug development research.

## 1. Introduction

Severe fever with thrombocytopenia syndrome (SFTS) virus (SFTSV), also called Dabie bandavirus, belongs to the genus *Bandavirus* in the family *Phenuiviridae*. It is an emerging tick-borne pathogen that is prevalent in China [1,2], South Korea [3], and Japan [4]. Major clinical symptoms of SFTSV infection include fever, gastrointestinal symptoms, hemorrhage, and consciousness deterioration. Total blood cell tests reveal thrombocytopenia and leukopenia [5]. Favipiravir (an antiviral drug) was reported as a potential drug for treating animals infected with SFTSV [6,7,8]. A clinical study to evaluate favipiravir’s efficacy in treating human patients with SFTS has been initiated in Japan [9]. SFTSV has three genomic segments of negative-sense or ambisense RNA, designated L, M, and S. The L-RNA segment encodes the viral RNA-dependent RNA polymerase (RdRp), the M-RNA segment encodes the glycoprotein precursor (GPC), and the S-RNA segment encodes the nucleoprotein (NP) and the nonstructural protein (NSs) [10].

Several approaches have screened for SFTSV replication inhibitors, including high-content screening to detect viral antigen and the use of recombinant SFTSVs expressing GFP or luciferase. However, any use of live SFTSV requires biosafety level (BSL) 3 (BSL3) containment, limiting opportunities for antiviral drug development. An alternative approach is developing high-throughput screening (HTS) assays based on critical viral functions that do not involve infectious materials.

SFTSV transcription and genome replication require two viral proteins, NP and RdRp. Because this RNA polymerase complex is essential for viral propagation, it is an attractive target for inhibitor development. A functional SFTSV RNA polymerase complex is reconstituted by transfecting the two components into mammalian cells. Its activity is measured through the coexpression of a model viral RNA that encodes a reporter gene flanked by the appropriate virus-derived cis-acting regulatory sequences [11,12]. Such systems are used to identify SFTSV RNA synthesis inhibitors. Chemical compound libraries are screened in BSL1 conditions.

This study optimized an SFTSV minigenome-based HTS assay in a 96-well format to identify small-molecule inhibitors of the SFTSV RNA synthesis machinery. A pilot screening of a bioactive library with 1594 compounds identified four compounds that inhibit SFTSV minigenome activity. The inhibitory chemical scaffolds included mycophenolate mofetil (MPM), methotrexate hydrate (MTX), clofarabine, and bleomycin (BLM). Overall, it was demonstrated that the SFTSV minigenome assay (MGA) is useful for anti-SFTSV drug development research and we identified the benzoquinoline series as broad-spectrum antivirals for SFTSV.

## 2. Materials and Methods

### 2.1. Cells and Drug Compounds

Human embryonic kidney HEK293T, human hepatocellular carcinoma Huh7, African green monkey kidney Vero, and hamster kidney BHK cells were cultured in Dulbecco’s modified Eagle medium (DMEM; Nacalai tesque, Kyoto, Japan) supplemented with 10% fetal bovine serum (FBS; Invitrogen, Waltham, MA, USA), 100 μg/mL of streptomycin, and 100 units of penicillin (Nacalai tesque). Favipiravir (6-fluoro-3-hydroxy-2-pyrazinecarboxamide) and ribavirin (1-b-D-ribofuranosyl-1,2,4-triazole-3-carboxamide) were obtained from Fujifilm Toyama Chemical Co. Ltd. (Toyama, Japan) and Fujifilm Wako Pure Chemical (Osaka, Japan), respectively. The compound libraries, namely FDA-approved Drug Library and G protein-coupled receptor (GPCR) Compound Library, were purchased from TargetMol (Boston, MA, USA). MMF, MTX, clofarabine, and bleomycin sulfate were purchased from Tokyo Chemical Industry (Tokyo, Japan). Thonzonium bromide, everolimus, and niclosamide were purchased from MedChemExpress (Monmouth Junction, NJ, USA).

### 2.2. Plasmid Construction

The cDNAs of the SFTSV SPL005A strain (GenBank: AB817990) were obtained by reverse transcription-PCR (RT-PCR) using a High Pure Viral RNA kit (Roche Applied Science, Upper Bavaria, Germany), SuperScript IV Reverse Transcriptase (Thermo Fisher Scientific, Waltham, MA, USA), and PrimeSTAR HS DNA Polymerase with GC Buffer (Takara, Shiga, Japan). We constructed an M segment-based minigenome system consisting of the expression of viral NP and RdRp proteins and a minigenome plasmid. The minigenome plasmid carried an enhanced green fluorescent protein (eGFP) with a HiBiT (HBT) peptide tag as a reporter gene (eGFPHBT). HBT is an 11-amino-acid peptide tag fused to the N or C terminus of the target protein or inserted into an accessible location within the protein structure. The amount of HBT-tagged protein expressed in a cell is determined by adding a lytic detection reagent containing the substrate, furimazine and Large BiT (LgBiT), the large subunit used in NanoLuc Binary Technology (NanoBiT). HBT binds tightly to LgBiT (K_D_ = 0.7 nM), forming a complex in the cell lysate that generates a bright, luminescent enzyme. The amount of luminescence is proportional to the amount of HBT-tagged protein in the cell lysate over seven orders of magnitude, with a glow-type luminescent signal stable for hours. This reporter gene was flanked by the viral untranslated regions (UTRs) in the viral genomic sense under the control of the human RNA polymerase I (pol I) promoter, and the human pol I terminator sequence. The viral M segment of SFTSV containing viral 3′ (leader) and 5′ (trailer) noncoding regions were amplified and cloned into the pPolIV plasmid vector, designated pPolIV-SFTSV-M. Next, all internal coding sequences were replaced with the coding sequence for an eGFP that is C-terminally tagged with HBT using NEBuilder HiFi DNA Assembly Master Mix (New England Biolabs, Ipswich, MA, USA), designated pPolIV-SFTSV-M-eGFPHBT. NP, RdRp, and NSs open reading frames were cloned into the expression vector, pKS336, under the human elongation factor-1α (HEF-1α) gene promoter [13]. The resulting plasmids were designated pKS-SFTSV-NP, pKS-SFTSV-L, and pKS-SFTSV-NSs for NP, RdRp, and NSs, respectively.

### 2.3. Minigenome Assay (MGA) and Screening Assay

To transfect the 293T cells in 12-well tissue culture plates, X-tremeGENE9 DNA Transfection Reagent (Roche Applied Science, Germany) was used following the manufacturer’s instructions. Each well was transfected with 200 ng pKS-SFTSV-NP, 100 ng pKS-SFTSV-L, and 200 ng pPolIV-SFTSV-M-eGFPHBT. The plasmids for transfection in 10 cm dishes were 2 μg pKS-SFTSV-NP, 1 μg pKS-SFTSV-L, and 2 μg pPolIV-SFTSV-M-eGFPHBT. One hour post transfection, cells were trypsinized and seeded in a 96-well tissue culture plate. The minigenome activity was assayed by measuring the luciferase activity at the indicated time points using the Nano-Glo HiBiT Lytic Detection System (Promega, Madison, WI, USA) following the manufacturer’s instructions. The luciferase signal was read using a GloMax Navigator Microplate Luminometer (Promega, Madison, WI, USA). The 50% inhibitory concentration (IC_50_) values were calculated using GraphPad Prism 7 (GraphPad Software, La Jolla, CA, USA) based on a four-parameter, nonlinear regression analysis. The test drug’s cytotoxicity was measured as described previously [6]. 293T cells were cultured for 2 d in the drug’s presence at designated concentrations with no plasmids. According to the manufacturer’s protocol, cell viability was measured using the RealTime Glo MT Cell Viability Assay.

## 3. Results

### 3.1. *Development* of a Pol I-Driven Minigenome System for SFTSV

We constructed an M segment-based SFTSV minigenome containing 3′ (leader) and 5′ (trailer) noncoding sequences. The internal coding sequence was replaced with eGFPHBT coding sequence (Figure 1). Protein expression plasmids under the control of the HEF-1α gene promoter allowed in trans expression of SFTSV, NP, and RdRp, which are the minimum requirements to initiate SFTSV replication and transcription [11]. The minigenome segment was cloned into a pol I expression plasmid in the negative orientation, which means that it must be first replicated by the NP plus RdRp replication complex before eGFP-HBT mRNA transcription can occur. In the presence of the pol I promoter, expressed NP and RdRp proteins can encapsidate the negative-sense reporter genome analog to generate artificial RNPs and can use viral UTRs as promoters of replication and transcription to drive reporter gene expression.

First, to evaluate the function of the minigenome constructs, we transfected 293T cells in a 12-well plate with a pol I-driven eGFP-HBT minigenome construct in the presence or absence of the necessary helper plasmids encoding NP and L. As shown in Figure 2A, no eGFP autofluorescence was seen with the minigenome in the absence of helper plasmids. However, in the presence of helper plasmids, cells exhibited robust eGFP autofluorescence 48 h after transfection. Cells transfected with the functional minigenome, but lacking the NP- or L-expressing plasmid, showed no eGFP signal. Next, we examined HBT luciferase activities in transfected 293T cells in a 12-well plate. As shown in Figure 2B, cells transfected with the functional minigenome, but lacking the NP- or L-expressing plasmid, showed a background activity comparable to that observed in mock-transfected 293T cells. However, in the presence of helper plasmids, cells exhibited more than 4000-fold minigenome activity than in cells transfected with only minigenome plasmid 48 h after transfection. These data showed that the expression plasmids of the viral N and L genes and minigenome systems were functional.

HBT luciferase activity in 293T cells was assessed at several time points post-transfection (Figure 3A). There was little activity 2 h post-transfection, but it increased to more than 4000-fold induction at 24 h post-transfection and peaked at 72 h post-transfection. We therefore chose 32 h post-transfection as the end-point of our SFTSV MGA due to the robust activity.

We next examined if the pol I minigenome system could be used in a broader range of cell lines, particularly those that are more difficult to transfect. Three cell lines from different species in the 12-well plate, including well-differentiated hepatocellular carcinoma-derived Huh7, African green monkey kidney-derived Vero, and hamster kidney-derived BHK, were transfected with the pol I-driven HBT luciferase minigenome components. At 32 h post-transfection, cells were lysed, and luciferase activity was measured (Figure 3B). The pol I minigenome systems worked well in Huh7 cells in a range approximately 3800-fold over a negative control. In contrast, the HBT luciferase activity was induced in a range of approximately 300-fold (Vero cells) to 30-fold (BHK cells) compared with the negative control. Since BHK cells are derived from a different species that is not human, pol I promoter hardly works, so it seems that the luciferase activity is lower than that of 293T and Huh7 cells. Therefore, we chose 293T cells for our SFTSV MGA because of their robust activity.

### 3.2. Augmentation of Minigenome Activity by NSs

It has been previously reported that Bunyamwera orthobunyavirus (BUNV), RVFV, and SFTSV (HB29 strain) NSs proteins inhibit the viral polymerase as measured by MGA [14,15]. To investigate whether the NSs of SFTSV SPL005A strain affect viral polymerase, M-segment-based minigenome plasmid (200 ng of pPolIV-SFTSV-M-eGFPHBT) was transfected into 293T cells in a 12-well plate along with 200 ng pKS-NP, 100 ng pKS-L, and increasing concentrations of pKS-NSs (up to 500 ng). As seen in Figure 3C, HBT luciferase activity was significantly increased when approximately 12.8 ng of pKS-NSs was added. However, HBT luciferase activity was inhibited when more than 32 ng of pKS-NSs was added. These results suggest that NSs protein coexpression with L and NP proteins substantially increases minigenome activities, although significant amounts showed marked inhibition. NSs forms distinct punctate structures in the cytoplasm of infected or plasmid-transfected cells. The NSs structures have been shown to colocalize with N protein and are associated with viral RNAs, suggesting that an appropriate amount of NSs plays a role in the replication of the virus.

### 3.3. Optimization of SFTSV Minigenome System for a 96-Well Format

We examined when to add inhibitors into the culture medium after transfection of three plasmids, pPolIV-SFTSV-M-eGFPHBT, pKS-NP, and pKS-L. We tested two small molecule inhibitors for activity against SFTSV minigenome activity. Two of these compounds, favipiravir and ribavirin, had been previously tested for activity against SFTSV and thus served as positive controls [6,16]. They have a broad spectrum of antiviral activity against various RNA viruses [17] and were previously described as SFTSV replication inhibitors [6,16]. Several times after transfection of 293T cells in 12-well plates, we trypsinized and seeded cells in 96-well plates in culture media with or without 640 mM of favipiravir or ribavirin. Following a 32 h incubation, HBT luciferase activity was assessed. As shown in Figure 4A, the inhibitory effects of favipiravir and ribavirin were found to be most effective after 1 h when compared at each time after transfection. Thus, we selected 1 h after transfection for trypsinizing and seeding cells in culture media with or without inhibitors after transfection.

To ensure that our minigenome system could be adapted to a high-throughput format, we first tested the system’s activity in 293T cells in a 96-well format. Three plasmids were transfected into 293T cells in 10 cm dishes to produce a uniform transfection of many cells. One hour post-transfection, cells were trypsinized and seeded in a 96-well plate in culture medium with or without increasing favipiravir or ribavirin concentration. After an additional 32 h incubation, HBT luciferase activity was assessed. As shown in Figure 4B, favipiravir and ribavirin inhibited SFTSV minigenome activity by approximately 2.5 log units at a concentration of 1000 mM. Favipiravir and ribavirin are potent inhibitors of SFTSV minigenome activity in a 96-well format. Favipiravir and ribavirin did not affect cell viability in the test range, as measured using the RealTime Glo MT Cell Viability Assay in parallel.

### 3.4. A High-Throughput Screen for Small Molecule Inhibitors of SFTSV Replication

Using the optimized conditions for a 96-well format, we screened a library of bioactive chemicals (1594 compounds) to identify compounds that inhibit SFTSV RNA synthesis. The optimized conditions included bulk-transfecting 293T cells in 10 cm dishes. The cells were then replated in 96-well plates in media containing inhibitors 1 h after transfection. A final concentration of 1% DMSO used in these assays did not affect minigenome activity (data not shown). We identified 171 compounds (10.7% of the total library) that inhibited minigenome activity by more than 50%, 22 (1.4% of the total library) of which reduced minigenome activity by more than 70% at a concentration of 5 μM. We retested them in triplicate in the minigenome assay and cytotoxicity assays. This identified seven compounds that inhibited the minigenome assay by at least 70% and exhibited less than 20% cell toxicity. From these, we selected four small molecule inhibitors for further study.

### 3.5. Validating Hit Compounds Identified in a Primary Screen

To evaluate the reproducibility of the identified hit compounds, seven compounds that varied in inhibition from 70% to 90% on the screen were chosen for retesting. The 50% inhibitory concentrations (IC_50_) and 50% cell cytotoxic concentrations (CC_50_) were assessed by measuring the reduction of substrate for the compounds in parallel to allow for the elimination of those that inhibited SFTSV minigenome activity by causing cell death (Table 1). The selectivity index (SI) of each compound was calculated using the CC_50_ and IC_50_ values (Table 1). Of the six compounds retested, we identified four compounds with IC_50_ values below their CC_50_ values, four of which have SI values of more than 500 (Figure 5 and Table 1). Hit compounds, mycophenolate mofetil, clofarabine, bleomycin, and methotrexate, had SI values of more than 500, suggesting that they inhibit SFTSV MGA at concentrations independent of their cytotoxic effects. Notably, these compounds did not reach a CC_50_ value when assayed at 32 h after addition in our assays. It is possible that some, such as the protein synthesis inhibitors, would have reached a CC_50_ value if the cytotoxicity assay was extended to 48 or 72 h after compound addition.

## 4. Discussion

Although some drugs have an inhibitory effect on SFTSV replication in vitro and in vivo [6], there are currently limited treatment options for SFTSV infection. This study described the establishment of an RNA polymerase I-based MGA for the recently emerged SFTSV and the use of this system as a HTS platform for potential antiviral drugs that interfere with viral transcription and replication processes.

Previously reported minigenome rescue systems for members of the family *Phleboviridae* have been based exclusively on a bacteriophage T7 polymerase-driven reverse genetics technology that requires the introduction of the exogenous T7 polymerase [10,18]. Although this can be achieved in several ways, including recombinant vaccinia virus infection and transient expression from plasmid DNA [19,20,21], this step can limit the system’s efficiency because of technical difficulties in achieving T7 expression in all cells. In contrast, pol I is a eukaryotic host cell polymerase which is localized in the nucleoli, transcribing mainly rRNA genes [22]. Using this enzyme to drive a reverse genetics system provides a substantial advantage because it alleviates the need to supply the polymerase in trans. Pol I localization in the nucleus is a potential limitation, especially for its application to viruses that replicate exclusively in the cytoplasm. However, it was previously employed successfully for the developing minigenome for several members of *Bunyaviridae* [23,24,25], *Arenaviridae* [26], and *Filoviridae* [27], which all replicate in the cytoplasm. Altogether, this indicates that pol I-driven transcription is a broadly applicable system for minigenome rescue, even for viruses that do not replicate in the nucleus.

Favipiravir is a pyrazine derivative, and after phosphoribosylation by host cellular kinases, metabolite favipiravir-RTP (favipiravir ribofuranosyl-5′-triphosphate) inhibits the activity of the RNA-dependent RNA polymerase of several viruses [17]. Ribavirin (1-b-D-ribofuranosyl-1,2,4-triazole-3-carboxamide) is a guanosine analog. It has a broad spectrum of activity against various RNA viruses through various mechanisms, including the reduction of viral RNA-dependent RNA polymerase activity, mutagenesis in the viral genome, inhibition of RNA capping, reduction of cellular inosine monophosphate dehydrogenase (IMPDH) activity, and modulation of the host immune response [28].

We identified four drug compounds that inhibited SFTSV replication at low micromolar concentrations, namely clofarabine, MPM, BLM, and MTX. Interestingly, clofarabine, MPM, and MTX are involved in the cellular nucleotide synthesis pathway. MPM, a prodrug form of mycophenolic acid (MPA), is an immunosuppressant, a noncompetitive, selective, and reversible inhibitor of IMPDH. IMPDH controls the rate of guanine monophosphate synthesis in the de novo pathway of purine synthesis, and is FDA-approved for immunosuppression. It has been reported that guanosine depletion by MPA inhibits viral RNA replication in HCV, chikungunya, and dengue viruses [29,30,31].

MTX, an FDA-approved folic acid antagonist and potent anti-inflammatory agent, inhibits dihydrofolate reductase (DHFR). It hinders intracellular folate metabolism and obstructs thymine and purine synthesis, prompting the hindrance of tumor development and cell death initiation through possible genotoxic effects or apoptosis [32]. Clofarabine is an FDA-approved ribonucleotide reductase (RNR) inhibitor used to treat acute lymphoblastic leukemia. As a purine nucleoside analog, clofarabine is transported into cells and is phosphorylated by host enzymes to the active forms of the drug, which subsequently prevents the formation of active enzyme. Owing to the necessity of RNR in de novo dNTP synthesis, this inhibition reduces endogenous dNTPs, which in turn inhibits DNA synthesis because of limited substrate availability [33]. BLM is an FDA-approved radiomimetic glycopeptide antitumor antibiotic produced by the bacterium *S. verticillus*. Its action mechanism causes single- and double-strand DNA breaks and initiates cleavage events on RNA and DNA molecules, interrupting the cell cycle. It has been used as an antineoplastic, especially for solid tumors [34]. Nucleoside analogs and inhibitors of cellular nucleos(t)ide synthesis pathways have frequently been identified as antiviral agents. MPA and MTX possess antiviral activity against a broad range of viruses. These reagents generally interfere with cellular nucleos(t)ide synthesis pathways, resulting in the depletion or imbalance of (d)NTP pools and activated innate immunity, thereby inducing the expression of interferon-stimulated genes, through nucleos(t)ide synthesis inhibition [35,36].

These findings could open up new opportunities for anti-SFTSV drug discovery. Notably, the benzoquinoline derivatives are novel and have not been reported as antiviral agents. Our study provides a foundation for the discovery of bioavailable derivatives for in vivo evaluation and for investigations into the mechanism of action of these antiviral agents.

## Figures and Tables

**Figure 1 viruses-13-01061-f001:**
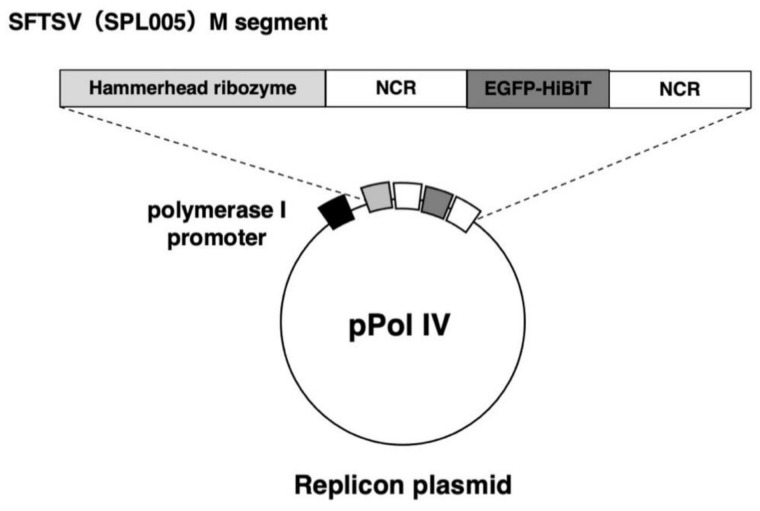
Schematic representation of cRNA-oriented pol I-driven minigenome. The reporter minigenome was composed of a reporter gene, eGFP, C-terminally tagged with HBT, flanked by the 3′ (leader) and 5′ (trailer) noncoding regions of SFTSV. They were flanked by hammerhead ribozyme (HamRz) sequence at the N-terminus. These transcription cassettes were then cloned between the transcriptional start and stop signals of the human RNA pol I.

**Figure 2 viruses-13-01061-f002:**
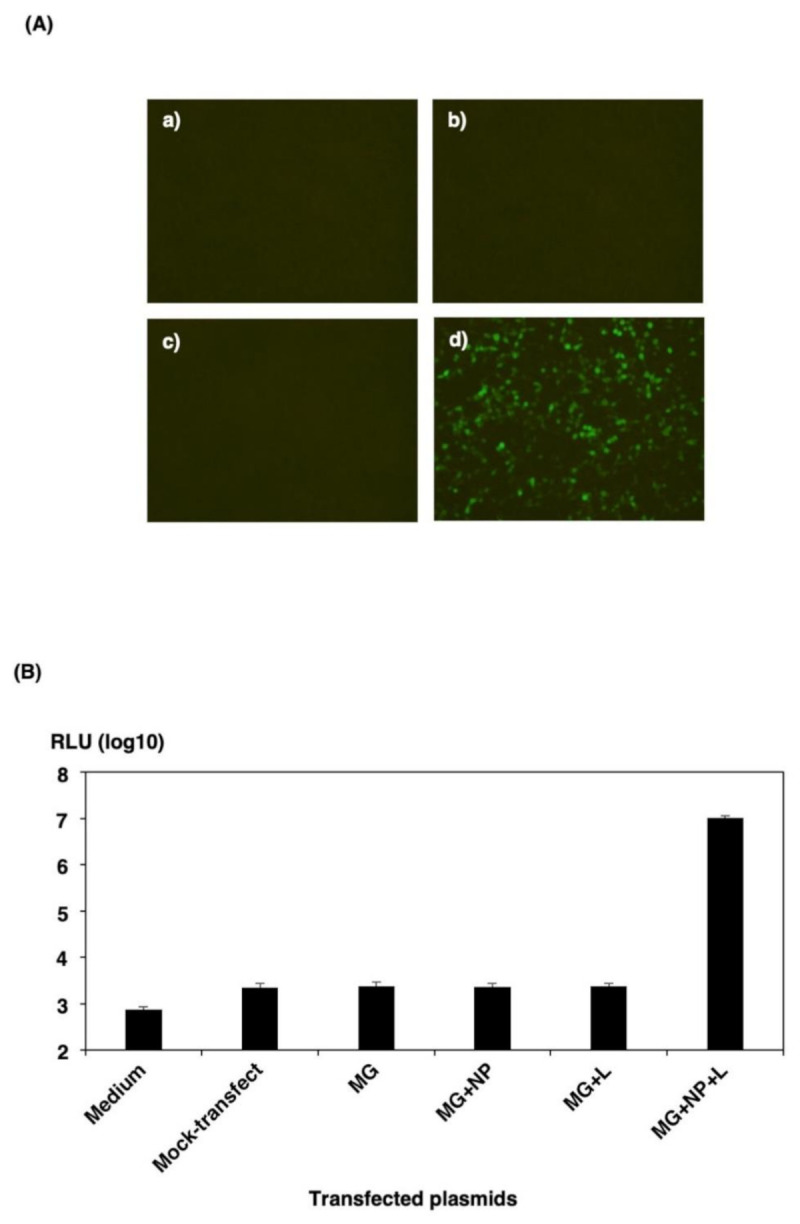
Reporter gene expression from cRNA-oriented pol I-driven minigenome. 293T cells in a 12-well plate were transfected, each with 300 ng of pPolIV-SFTSV-M-eGFPHBT (MG), 300 ng of pKS-SFTSV-NP, and 150 ng of pKS-SFTSV-L plasmid per well. (**A**) 293T cells were transfected with the MG (**a**), MG and pKS-SFTSV-NP (**b**), MG and pKS-SFTSV-L (**c**), or MG, pKS-SFTSV-NP, and pKS-SFTSV-L (**d**) and analyzed 48 h post-transfection for expression of eGFP-HBT by fluorescence microscopy. (**B**) 293T cells were either transfected with empty vector (mock-transfection) or transfected with the MG, MG and pKS-SFTSV-NP (MG + NP), MG and pKS-SFTSV-L (MG + L), or MG, pKS-SFTSV-NP, and pKS-SFTSV-L (MG + NP + L). The minigenome activity of eGFP-HBT expression was assayed by measuring the luciferase activity 32 h post-transfection using the Nano-Glo HiBiT Lytic Detection System. The results shown are for three independent assays, with error bars representing standard deviations.

**Figure 3 viruses-13-01061-f003:**
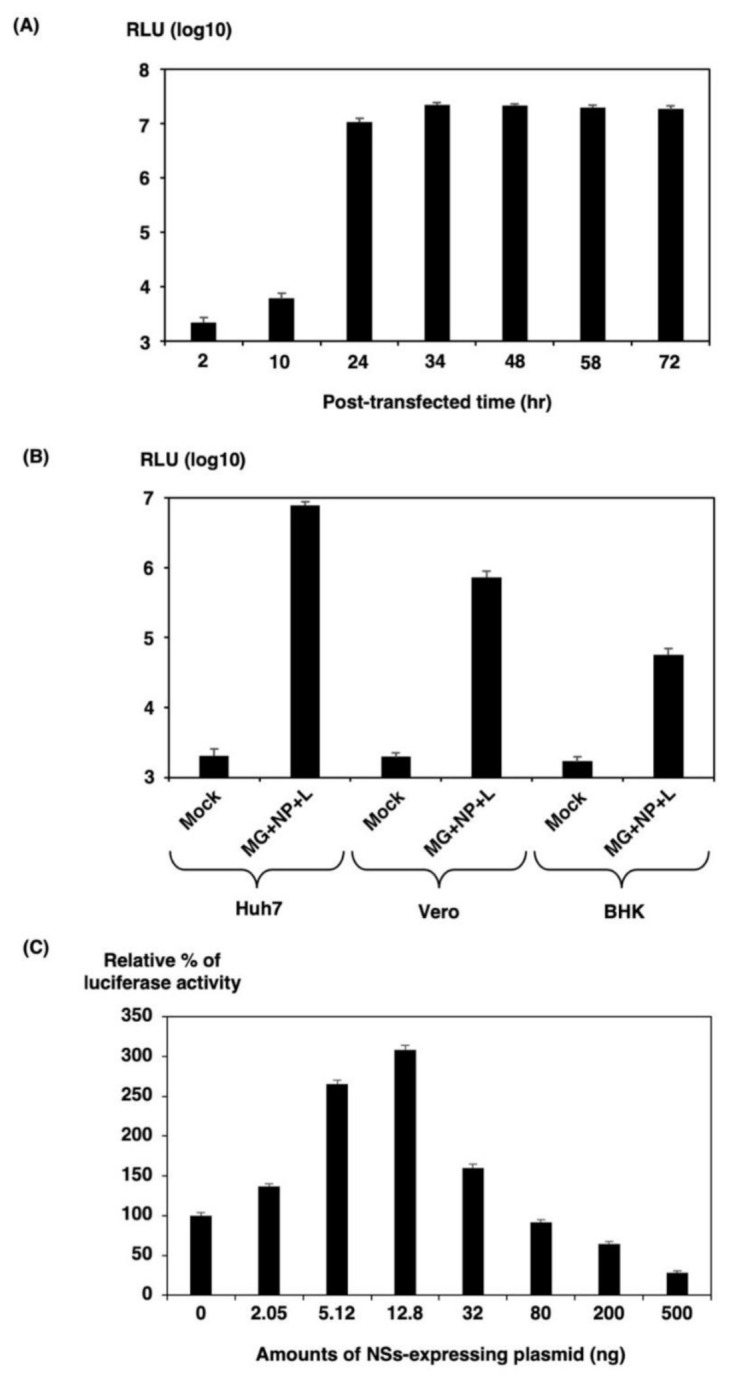
293T cells in a 12-well plate were transfected with 300 ng of MG plasmid along with the necessary supporting plasmids (300 ng of NP and 1500 ng of pKS-SFTSV-L) per well. (**A**) Time course of minigenome activity. At each time point post-transfection, luciferase activity was assessed. (**B**) Comparison of the minigenome activity in the three cell lines. Huh7, Vero, or BHK cells in 12-well plates were transfected with the indicated plasmids. Empty pKS336 vector was used to ensure that the total amount of DNA used in each transfection had the same minigenome activity of eGFP-HBT expression and was assayed by measuring the luciferase activity 32 h post-transfection using the Nano-Glo HiBiT Lytic Detection System (Promega, Madison, WI). (**C**) Effect of NSs protein on minigenome activity. 293T cells were transfected with MG, pKS-SFTSV-NP, and pKS-SFTSV-L, indicating increasing amounts of pKS-NSs or pKS336. Empty vector pKS336 was used to ensure that equal amounts of DNA were transfected into each reaction mixture. The minigenome activity was assayed by measuring the luciferase activity 32 h post-transfection. The minigenome activity was expressed as the fold induction of normalized luciferase units relative to the background control (absence of NS-expressing plasmid), corrected as a percentage of the activity observed in the presence of NP and L proteins for each MG. The results shown are for three independent assays, with error bars representing standard deviations.

**Figure 4 viruses-13-01061-f004:**
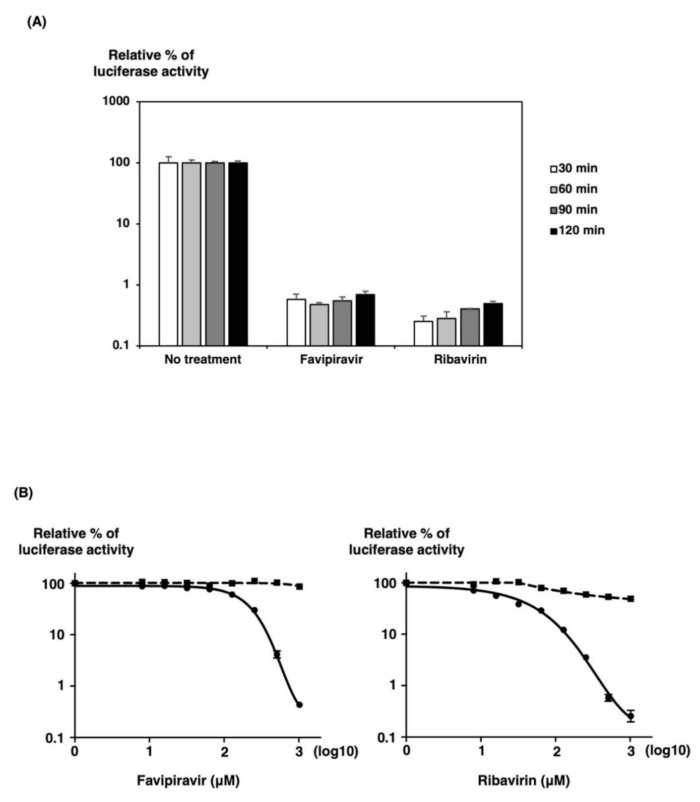
The SFTSV MG assay for favipiravir or ribavirin. (**A**) 293T cells were transfected with MG plasmid and the necessary supporting plasmids (pKS-SFTSV-NP and pKS-SFTSV-L). At each time point post-transfection, 293T cells were seeded in a 96-well plate in media containing 640 mM of favipiravir or ribavirin. After 32 h of incubation, luciferase activities were measured. (**B**) Inhibitory effect of favipiravir and ribavirin on SFTSV MG activity (●) in 293T cells. One hour after transfection with MG, pKS-SFTSV-NP, and pKS-SFTSV-L plasmids, 293T cells were seeded in a 96-well plate in media containing various concentrations of favipiravir or ribavirin. After 32 h of incubation, luciferase activities were measured. MG activity is expressed as the fold induction of normalized luciferase units relative to the background control (absence of reagents), corrected as a percentage of the activity observed in the presence of NP and L proteins for each MG. Cell viability (◼) was determined in the presence of drugs using the RealTime Glo MT Cell Viability Assay in parallel. A sigmoidal dose–response curve was fitted to the data using GraphPad Prism 7 (GraphPad Software, San Diego, CA, USA). The results shown are for three independent assays, with error bars representing standard deviations.

**Figure 5 viruses-13-01061-f005:**
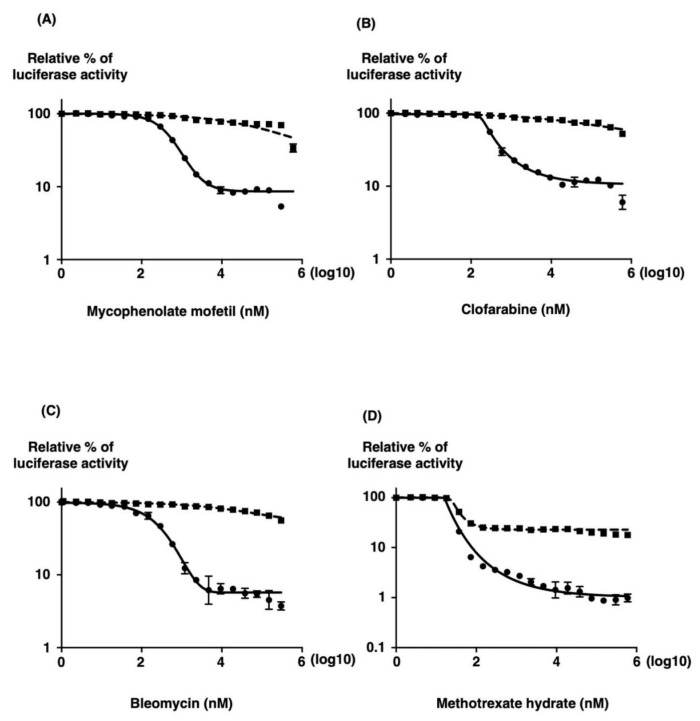
Inhibitory effect of mycophenolate mofetil, clofarabine, bleomycin, and methotrexate hydrate on SFTSV MG activity (●) in 293T cells. (**A**) One hour after transfection with MG, pKS-SFTSV-NP, and pKS-SFTSV-L plasmids, 293T cells were seeded in a 96-well plate in media containing various concentrations of mycophenolate mofetil (**A**), clofarabine (**B**), bleomycin (**C**), and methotrexate hydrate (MTX) (**D**). After 32 h of incubation, luciferase activities were measured. Cell viability (◼) was determined in the presence of drugs using the RealTime Glo MT Cell Viability Assay (Promega, Madison, WI) in parallel. The minigenome activity is expressed as the fold induction of normalized luciferase units relative to the background control (absence of reagents), corrected as a percentage of the activity observed in the presence of NP and L proteins for each MG. A sigmoidal dose–response curve was fitted to the data using GraphPad Prism 7. The results shown are for three independent assays, with error bars representing standard deviations.

**Table 1 viruses-13-01061-t001:** Anti-SFTSV activity and cytotoxicity of six selected drug compounds.

Compound	% Inhibition in Screen	CC_50_ (μM)	IC_50_ (nM)	SI (CC_50_/IC_50_)
Mycophenolate mofetil	72.0	500	580	862
Clofarabine	80.0	>600	580	>1034
Bleomycin	90.9	>300	290	>1034
Methotrexate hydrate	90.4	70	140	500
Thonzonium	83.3	8.6	2000	4.3
Everolimus	75.0	11.3	1000	11.3

## Data Availability

Not applicable.

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
