# Peer review of "M Segment-Based Minigenome System of Severe Fever with Thrombocytopenia Syndrome Virus as a Tool for Antiviral Drug Screening"

_viruses, 2021, doi:10.3390/v13061061_

Round 1

Reviewer 1 Report

This manuscript describes a useful SFTSV mini-genome system analogous to several similar systems for key species in the Bunyavirales. The authors provide convincing data, which indicate that this mini-genome system is useful for studying basic facets of virus replication, and likely also utility as a screening tool for anti-viral therapeutics targeting the polymerase. Overall, I think this is a good contribution to the field and a worthwhile paper. The authors use the mini-genome system to screen a small molecule library to demonstrate utility. The resulting data indicate that the system works. However, the extent of inhibition is rather modest (see comment 1 below).

Comments:

  1. Rather than trying to make too much of the limited inhibition of several of the small molecules that are considered hits in the screen, I think the authors would provide a service to the field by pointing out that there is ample space for identifying more robust polymerase inhibitors against this and other related viruses. At least pointing out that none of these small molecules markedly affect polymerase activity. For that matter, the T-705 and ribavirin positive controls may not actually be all that robust (see comments 7 and 8 below)
  2. Isn’t also a possibility that inhibition by small molecules in the library is actually mediated through targeting of NP (or even the viral RNA directly) rather than through pol?
  3. The luc component of the GFP-HBT reporter, which is described in the Discussion, should be described more completely earlier in the manuscript (either the Methods section or an initial part of the Results. While GFP is well known, many readers will not know that HiBiT is one component of a part of a composite assay that can generate luc activity and will likely be confused when they first look at presented data.
  4. Line 150: “N-terminals” should be N-terminus
  5. It’s interesting that the reporter and mini-genome system worked with such different efficiency in different cell types. Discussion is warranted.
  6. The experiment of Fig. 3C is interesting since it shows that NSs can positively or negatively affect replication depending on relative amount. The experiment doesn’t determine whether this effect specifically involves targeting of pol by NSs. Worth also adding a paragraph on this to the discussion.
  7. Line 225: The authors state that Favipiravir and ribavirin are potent inhibitors of minigenome activity. 1000 uM is a pretty high concentration. It’s fair to say that these inhibitors were useful for demonstrating that the mini-genome system works but not that these known inhibitors are robust (doesn’t take away from the system itself but that the IC50 of favipiravir and ribavirin may be pretty high for SFTSV pol).
  8. 2B demonstrates that the mini-genome system works very well, is NP and L dependent, and that there’s about a 4-log amplification of the genome above background under optimal conditions. T-705, ribavirin, and MTX inhibit replication by about 2 logs, and that CLF, MTX, and BLM inhibit replication by 1 log.
  9. Graphs throughout. The authors should add more complete x and y labels.

Author Response

Reviewer 1

This manuscript describes a useful SFTSV mini-genome system analogous to several similar systems for key species in the Bunyavirales. The authors provide convincing data, which indicate that this mini-genome system is useful for studying basic facets of virus replication, and likely also utility as a screening tool for anti-viral therapeutics targeting the polymerase. Overall, I think this is a good contribution to the field and a worthwhile paper. The authors use the mini-genome system to screen a small molecule library to demonstrate utility. The resulting data indicate that the system works. However, the extent of inhibition is rather modest (see comment 1 below).

>Thank you very much for summarizing our manuscript. As you can say, use of mini-genome system is useful for studying viral replication. Especially in SFTSV study, which must be handled at BSL3 facilities, we believe that it can be used more easily and effectively.

Comments:

  1. Rather than trying to make too much of the limited inhibition of several of the small molecules that are considered hits in the screen, I think the authors would provide a service to the field by pointing out that there is ample space for identifying more robust polymerase inhibitors against this and other related viruses. At least pointing out that none of these small molecules markedly affect polymerase activity. For that matter, the T-705 and ribavirin positive controls may not actually be all that robust (see comments 7 and 8 below)

>Even favipiravir and ribavirin may appear to be suppressed only at high concentrations, as the replicon assay is less sensitive than the live virus assay. However, since replicon assay is a simple and safe system, we think it can be used as a guide for antiviral screening.

  1. Isn’t also a possibility that inhibition by small molecules in the library is actually mediated through targeting of NP (or even the viral RNA directly) rather than through pol?

>We agree with your suggestion. It is quite possible that inhibition by small molecules is targeted to the NP rather than the polymerase.

  1. The luc component of the GFP-HBT reporter, which is described in the Discussion, should be described more completely earlier in the manuscript (either the Methods section or an initial part of the Results. While GFP is well known, many readers will not know that HiBiT is one component of a part of a composite assay that can generate luc activity and will likely be confused when they first look at presented data.

>Thank you for your suggestion. We moved to the description of HiBiT from the Discussion section to the Methods section in the revised manuscript (Line 92-100).

  1. Line 150: “N-terminals” should be N-terminus

>Thank you for your suggestion. We mistook it. We changed it in the revised manuscript (Line 150).

  1. It’s interesting that the reporter and mini-genome system worked with such different efficiency in different cell types. Discussion is warranted.

>Thank you for your suggestion. We added the description “Since BHK cells are derived from a different species that is not human, pol I promoter hardly works, so it seems that the luciferase activity is lower than that of 293T and Huh7 cells.” in the revised manuscript (Line 184-185).

  1. The experiment of Fig. 3C is interesting since it shows that NSs can positively or negatively affect replication depending on relative amount. The experiment doesn’t determine whether this effect specifically involves targeting of pol by NSs. Worth also adding a paragraph on this to the discussion.

>Thank you for your suggestion. We added the description “NSs forms distinct punctate structures in the cytoplasm of infected or plasmid-transfected cells. The NSs structures have been shown to colocalize with N protein and to be associated with viral RNAs, suggesting that appropriate amount of NSs plays a role in the replication of the virus.” in the revised manuscript (Line 213-216).

  1. Line 225: The authors state that Favipiravir and ribavirin are potent inhibitors of minigenome activity. 1000 uM is a pretty high concentration. It’s fair to say that these inhibitors were useful for demonstrating that the mini-genome system works but not that these known inhibitors are robust (doesn’t take away from the system itself but that the IC50 of favipiravir and ribavirin may be pretty high for SFTSV pol).

>Thank you for your comment. The 1000uM concentration of a drug is generally very high, as you said, but CC50 of ribavirin or favipiravir is known to at 4000uM or more. Therefore, even 1000uM is not so toxic. The inhibitory effects of ribavirin or favipiravir for the live virus are higher than those for the replicon. This is because the amount of replication of the replicon is lower than that of the live virus.

  1. 2B demonstrates that the mini-genome system works very well, is NP and L dependent, and that there’s about a 4-log amplification of the genome above background under optimal conditions. T-705, ribavirin, and MTX inhibit replication by about 2 logs, and that CLF, MTX, and BLM inhibit replication by 1 log.

>The effect of replication due to the presence of NP and L shows 4 logs or more, while the inhibitory effect of the drugs are 1 to 2 logs, which is considered to be the limit of the effects.

  1. Graphs throughout. The authors should add more complete x and y labels.

>Thank you for your suggestion. We described additional x and y labels in all the graphs in the revised manuscript. 

Reviewer 2 Report

The authors performed a high-throughput screening of small molecules that can potentially inhibit RNA synthesis, via targeting RdRp, of the tick-borne pathogen SFTSV. The authors used a minigenome assay in a 96-well plate to identify applicable candidates. The assay resulted in 4 novel candidates in addition to identifying already clinically-approved inhibitors (that were used as positive controls). Subsequent small-scale screening of these 4 candidates inhibited >80% SFTSV activity with <20% cytotoxicity. The authors propose that a minigenome assay is useful in determining anti-SFTSV drugs.

Although the methods and results are sound, the authors need to improve the English. Redundancy, run-on sentences and poorly structured sentences are of main concern - aside from small grammatical errors. These comments are examples to assist the authors in correcting the manuscript's syntax:

Abstract

Line 24: change "... was screened using a high-throughput screening..." to, for example, "... were processed using..."

Line 27: "...favipiravir, and ribavirin. " there's no comma before 'and'

Introduction

Lines 48-49: change "Several approaches have been taken to screen for inhibitors of SFTSV replication inhibitors, ..." to "Several approaches have screened for SFTSV replication inhibitors, ..."

Lines 55-60: the excessive use of "can be" can be reduced for a better syntax of the paragraph.

Lines 66-68: Poorly written phrase. Re-word

Materials and Methods

Line 77: change "was" to "were"

Line 91: "HiBiT" font is larger than the rest

Line 101: "... 1a (HEF-1a) ..." font is larger than the rest

Results

Lines 124-125: "... HEF-1a..." font is larger than the rest

Lines 138-139: a comma before "but" and after "plasmid"

Line 141: a comma before "but" and after "plasmid"

Lines 163-165: a run-on sentence

Line 165: no comma before-after "therefore"

Lines 206-207: remove "of" from "We tested two of small..."

Lines 214-215: Re-word sentence

Lines 264-267: Re-word; the use of a semi-colon is also not necessary and should be two separate sentences

Discussion

Line 336: "… S. verticillus." font is larger than the rest

Additional comment

All the above suggestions should assist the authors in revising, overall, the Discussion - for which the syntax can be improved to better convey relations with the Results.

Author Response

Reviewer 2

The authors performed a high-throughput screening of small molecules that can potentially inhibit RNA synthesis, via targeting RdRp, of the tick-borne pathogen SFTSV. The authors used a minigenome assay in a 96-well plate to identify applicable candidates. The assay resulted in 4 novel candidates in addition to identifying already clinically-approved inhibitors (that were used as positive controls). Subsequent small-scale screening of these 4 candidates inhibited >80% SFTSV activity with <20% cytotoxicity. The authors propose that a minigenome assay is useful in determining anti-SFTSV drugs.

Although the methods and results are sound, the authors need to improve the English. Redundancy, run-on sentences and poorly structured sentences are of main concern - aside from small grammatical errors. These comments are examples to assist the authors in correcting the manuscript's syntax:

>Thank you for your comments. Please forgive our writing skills for expressing in English.

Abstract

Line 24: change "... was screened using a high-throughput screening..." to, for example, "... were processed using..."

Line 27: "...favipiravir, and ribavirin. " there's no comma before 'and'

> We changed them in the revised manuscript following your suggestion.  

Introduction

Lines 48-49: change "Several approaches have been taken to screen for inhibitors of SFTSV replication inhibitors, ..." to "Several approaches have screened for SFTSV replication inhibitors, ..."

Lines 55-60: the excessive use of "can be" can be reduced for a better syntax of the paragraph.

> We changed them in the revised manuscript following your suggestion.

Lines 66-68: Poorly written phrase. Re-word

> We changed the description of “.. it was demonstrated that the SFTSV minigenome assay (MGA) is useful for anti-SFTSV drug development research, and was identified...” in lines 67-68 in the revised manuscript.

Materials and Methods

Line 77: change "was" to "were"

Line 91: "HiBiT" font is larger than the rest

Line 101: "... 1a (HEF-1a) ..." font is larger than the rest

> We corrected them in the revised manuscript following your suggestion.

Results

Lines 124-125: "... HEF-1a..." font is larger than the rest

Lines 138-139: a comma before "but" and after "plasmid"

Line 141: a comma before "but" and after "plasmid"

> We corrected them in the revised manuscript following your suggestion.

Lines 163-165: a run-on sentence

> We changed the description of “There was little activity 2 h post-transfection, but increased to more than 4,000-fold induction at 24 h post-transfection and peaked at 72 h post-transfection.” in lines 171-172 in the revised manuscript.

Line 165: no comma before-after "therefore"

Lines 206-207: remove "of" from "We tested two of small..."

> We changed them in the revised manuscript following your suggestion.

Lines 214-215: Re-word sentence

> We changed the description of “As shown in Fig. 4A, the inhibitory effects of favipiravir and ribavirin were found to be most effective after 1 hour when compared at each time after transfection.” in lines 222-223 in the revised manuscript.

Lines 264-267: Re-word; the use of a semi-colon is also not necessary and should be two separate sentences

 > We changed it in the revised manuscript following your suggestion.

Discussion

Line 336: "… S. verticillus." font is larger than the rest

 > We corrected it in the revised manuscript following your suggestion.

Reviewer 3 Report

Yamada et al., generated an SFTSV minigenome assay (MGA), screened an FDA approved drug library in 96-well plates, and identified four potential candidates against SFTSV. However, the novelty of this study is low. The use of the human RNA polymerase I promoter, compared to T7 RNA promoter in two previous studies and minigenome systems of all other RNA viruses, does not make profound significance, since exogenous DNA plasmid template is anyway not existing in the authentic replication of SFTSV. Furthermore, the operation of the MGA assay is also not simpler than the previous ones. Last but not least, the four candidates identified were not tested in in vitro infection systems, in which the screening of FDA approved drugs were already performed and published.

Comments:

  1. I don’t think that the use of human RNA polymerase I promoter makes a significance compared to the T7 RNA promoter in the plasmid, because exogenous DNA template for RNA transcription is anyway not existing in the genuine replication of SFTSV, as a RNA virus without retro transcription activity. The authors suggested that T7 polymerase-driven reverse genetics requires an introduction of the exogenous T7 polymerase. However, a stable cell line BSR-T7/5 expressing the T7 RNA polymerase was generated and widely used in the field (Brennan et al, J Virol, 2015; Rezelj et al., J Virol, 2019, etc.). Such stable cells expressing T7 polymerase have also been used in the Hepatitis C virus replicon and other systems. In this concept the MGA system shown here has low novelty unless the authors can consider another system, for instance, delivering in vitro transcribed RNA into the cells.
  2. The authors concluded that the MGA assay lowers the requirement of a biosafety facility. However, this has also been solved by introducing virus-like particles and pseudovirus expressing real SFSTV glycoproteins and encoding a reporter gene.
  3. In this study, four candidates were identified by the MGA assay. However, whether they are functional was never evaluated using an in vitro infection assay. Without the data, it is difficult to conclude whether they are real candidates for further development. Actually similar screening of an FDA approved drug library has also been performed using real SFTSV virions (Li et al., Cell Res, 2019), which is advanced than any other minigenome system. I am wondering why the four candidates were not identified there.
  4. Transient co-transfection of three plasmids is not stable when the experiments are repeated, and usually i.e. the other luciferase is required for normalization of transfection efficiency. In this study, neither single transfection of pPol-SFTSV-M-eGFPHBT plasmid (because the authors could generate a cell line with stable NP and L protein expression) nor a normalization method was shown. For instance, how can we know transfection efficiency and the number of cells prior to measurement are the same in figures 2 and 3?
  5. Nucleoside analogs have a broad antiviral activity, especially for RNA viruses. They are likely not specific for SFTSV. It holds similar to favipiravir and ribavirin. For those compounds, the 640 uM concentration used is incredibly high, indicating that the system was not sensitive enough. How to explain the fact that favipiravir had an IC50 value of 6 uM in SFTSV infection (Tani et al., mSphere, 2016) but it showed an IC50 value of >100 uM as shown in figure 4B?

Author Response

Reviewer 3

Yamada et al., generated an SFTSV minigenome assay (MGA), screened an FDA approved drug library in 96-well plates, and identified four potential candidates against SFTSV. However, the novelty of this study is low. The use of the human RNA polymerase I promoter, compared to T7 RNA promoter in two previous studies and minigenome systems of all other RNA viruses, does not make profound significance, since exogenous DNA plasmid template is anyway not existing in the authentic replication of SFTSV. Furthermore, the operation of the MGA assay is also not simpler than the previous ones. Last but not least, the four candidates identified were not tested in in vitro infection systems, in which the screening of FDA approved drugs were already performed and published.

>Thank you for your comments. We are currently preparing an experimental system using a live virus (including the development of BSL3 experimental facility). It will soon be possible to evaluate the antiviral effects of the four candidate drugs on SFTSV.

Comments:

  1. I don’t think that the use of human RNA polymerase I promoter makes a significance compared to the T7 RNA promoter in the plasmid, because exogenous DNA template for RNA transcription is anyway not existing in the genuine replication of SFTSV, as a RNA virus without retro transcription activity. The authors suggested that T7 polymerase-driven reverse genetics requires an introduction of the exogenous T7 polymerase. However, a stable cell line BSR-T7/5 expressing the T7 RNA polymerase was generated and widely used in the field (Brennan et al, J Virol, 2015; Rezelj et al., J Virol, 2019, etc.). Such stable cells expressing T7 polymerase have also been used in the Hepatitis C virus replicon and other systems. In this concept the MGA system shown here has low novelty unless the authors can consider another system, for instance, delivering in vitro transcribed RNA into the cells.

>As you say, the rescue system using T7 polymerase has been applied by many other viruses, and then this system should be fine. However, in our experiment, the system using T7 polymerase did not work for some reason, so we utilized the rescue system using human RNA polymerase I promoter.

  1. The authors concluded that the MGA assay lowers the requirement of a biosafety facility. However, this has also been solved by introducing virus-like particles and pseudovirus expressing real SFSTV glycoproteins and encoding a reporter gene.

>As you say, pseudotyped virus can also be used with reduced biosafety levels. however, the pseudotyped virus can only analyze for the entry steps of infection. On the other hand, MGA system can only analyze for the replication steps of infection. It is better to search for antiviral drugs with different mechanisms of action in each system.

  1. In this study, four candidates were identified by the MGA assay. However, whether they are functional was never evaluated using an in vitro infection assay. Without the data, it is difficult to conclude whether they are real candidates for further development. Actually similar screening of an FDA approved drug library has also been performed using real SFTSV virions (Li et al., Cell Res, 2019), which is advanced than any other minigenome system. I am wondering why the four candidates were not identified there.

>The selection of candidate drugs may differ depending on the cell type or an assay method. Soon we will also evaluate using a live virus.

  1. Transient co-transfection of three plasmids is not stable when the experiments are repeated, and usually i.e. the other luciferase is required for normalization of transfection efficiency. In this study, neither single transfection of pPol-SFTSV-M-eGFPHBT plasmid (because the authors could generate a cell line with stable NP and L protein expression) nor a normalization method was shown. For instance, how can we know transfection efficiency and the number of cells prior to measurement are the same in figures 2 and 3?

>Transfection efficiency is significantly affected when compared with different cell types and may require internal standardization. We believe that different plasmids will not affect transfection efficiency if transfected into the same cell type. In Fig. 3B, no internal standardization is required for the experiment to find the most efficient cell type to use for screening.

  1. Nucleoside analogs have a broad antiviral activity, especially for RNA viruses. They are likely not specific for SFTSV. It holds similar to favipiravir and ribavirin. For those compounds, the 640 uM concentration used is incredibly high, indicating that the system was not sensitive enough. How to explain the fact that favipiravir had an IC50 value of 6 uM in SFTSV infection (Tani et al., mSphere, 2016) but it showed an IC50 value of >100 uM as shown in figure 4B?

>As you say, the evaluation of drug concentration with replicon assay is less active than the live virus assay. This is because the amount of live virus replication is much higher than that of replicon. In replicon, the amount of replication is limited to the amount of protein supplied from the plasmid, which reduces the amount of replication. So, we think this is the limit of this assay system.

Round 2

Reviewer 3 Report

Rebuttal Letter:

"Yamada et al., generated an SFTSV minigenome assay (MGA), screened an FDA approved drug library in 96-well plates, and identified four potential candidates against SFTSV. However, the novelty of this study is low. The use of the human RNA polymerase I promoter, compared to T7 RNA promoter in two previous studies and minigenome systems of all other RNA viruses, does not make profound significance, since exogenous DNA plasmid template is anyway not existing in the authentic replication of SFTSV. Furthermore, the operation of the MGA assay is also not simpler than the previous ones. Last but not least, the four candidates identified were not tested in in vitro infection systems, in which the screening of FDA approved drugs were already performed and published.

>>Thank you for your comments. We are currently preparing an experimental system using a live virus (including the development of BSL3 experimental facility). It will soon be possible to evaluate the antiviral effects of the four candidate drugs on SFTSV."

I accept your argument and agree that establishing a BSL3 facility will take time. Although infection with a live virus is currently not available, it is very disappointing that the authors did not show any experimental evidence over a five-week revision to defense this new system is reliable. It is common sense that any new system/method has to be compared with old-established ones tested in many laboratories. Over a month, the authors shall provide biochemical (e.g. T7 promoter versus that in the manuscript) and cell culture data, which can be performed in BSL1 and BSL2 facilities. 

Comments:

  1. "I don’t think that the use of human RNA polymerase I promoter makes a significance compared to the T7 RNA promoter in the plasmid, because exogenous DNA template for RNA transcription is anyway not existing in the genuine replication of SFTSV, as a RNA virus without retro transcription activity. The authors suggested that T7 polymerase-driven reverse genetics requires an introduction of the exogenous T7 polymerase. However, a stable cell line BSR-T7/5 expressing the T7 RNA polymerase was generated and widely used in the field (Brennan et al, J Virol, 2015; Rezelj et al., J Virol, 2019, etc.). Such stable cells expressing T7 polymerase have also been used in the Hepatitis C virus replicon and other systems. In this concept the MGA system shown here has low novelty unless the authors can consider another system, for instance, delivering in vitro transcribed RNA into the cells.

>>As you say, the rescue system using T7 polymerase has been applied by many other viruses, and then this system should be fine. However, in our experiment, the system using T7 polymerase did not work for some reason, so we utilized the rescue system using human RNA polymerase I promoter."

As far as I know, T7 RNA promoter is rather short in length (5′ TAATACGACTCACTATAG 3′). I do not believe that within five weeks the author could not finish molecular cloning and further compare T7 RNA promoter with RNA promoter. The authors mentioned "the system using T7 polymerase did not work for some reason" which is not scientific. A negative result is a result, yet not trying and showing this result that helps understanding the difference between T7 and human RNA polymerase I promoter is not acceptable. Besides, the authors did not cite any other systems of SFTSV in the manuscript. How can they conclude this new system superior?

  1. "The authors concluded that the MGA assay lowers the requirement of a biosafety facility. However, this has also been solved by introducing virus-like particles and pseudovirus expressing real SFSTV glycoproteins and encoding a reporter gene.

>>As you say, pseudotyped virus can also be used with reduced biosafety levels. however, the pseudotyped virus can only analyze for the entry steps of infection. On the other hand, MGA system can only analyze for the replication steps of infection. It is better to search for antiviral drugs with different mechanisms of action in each system."

This is a nice answer. Your system is showing merit since other systems like VLPs can solely be used in studying viral entry.

  1. "In this study, four candidates were identified by the MGA assay. However, whether they are functional was never evaluated using an in vitro infection assay. Without the data, it is difficult to conclude whether they are real candidates for further development. Actually similar screening of an FDA approved drug library has also been performed using real SFTSV virions (Li et al., Cell Res, 2019), which is advanced than any other minigenome system. I am wondering why the four candidates were not identified there.

>>The selection of candidate drugs may differ depending on the cell type or an assay method. Soon we will also evaluate using a live virus.".

The reply is not persuasive, because once using the FDA-approved drug library, it is common to get "hits" in many other studies. In the manuscripts the authors ever tried two inhibitors favipiravir and ribavirin that are known to inhibit SFTSV, however, their IC50 values were pretty high. As answered, if it is a cell-type-specific effect, the authors during the five-week revision should test the four inhibitors using more susceptible cell lines for SFTSV infection. There is no further information for the evaluation.

  1. "Transient co-transfection of three plasmids is not stable when the experiments are repeated, and usually i.e. the other luciferase is required for normalization of transfection efficiency. In this study, neither single transfection of pPol-SFTSV-M-eGFPHBT plasmid (because the authors could generate a cell line with stable NP and L protein expression) nor a normalization method was shown. For instance, how can we know transfection efficiency and the number of cells prior to measurement are the same in figures 2 and 3?

>Transfection efficiency is significantly affected when compared with different cell types and may require internal standardization. We believe that different plasmids will not affect transfection efficiency if transfected into the same cell type. In Fig. 3B, no internal standardization is required for the experiment to find the most efficient cell type to use for screening."

I was not arguing the transfection efficiency among cell types, but the different efficiencies when you transfected the same cell type in different experiments. It is a chemical transfection of three plasmids. How can you make sure the amount ratio of three plasmids (w/w/w) are always the same.

In Figure 3B, assuming that you have seeded the same number of cells before transfection, without a normalizer how to conclude that the cell numbers are still the same during measurement due to cytotoxicity induced by the transfection reagent?

  1. "Nucleoside analogs have a broad antiviral activity, especially for RNA viruses. They are likely not specific for SFTSV. It holds similar to favipiravir and ribavirin. For those compounds, the 640 uM concentration used is incredibly high, indicating that the system was not sensitive enough. How to explain the fact that favipiravir had an IC50 value of 6 uM in SFTSV infection (Tani et al., mSphere, 2016) but it showed an IC50 value of >100 uM as shown in figure 4B?

>As you say, the evaluation of drug concentration with replicon assay is less active than the live virus assay. This is because the amount of live virus replication is much higher than that of replicon. In replicon, the amount of replication is limited to the amount of protein supplied from the plasmid, which reduces the amount of replication. So, we think this is the limit of this assay system."

This answer is accepted but should be present in the discussion.